# Amyloids as Building Blocks for Macroscopic Functional Materials: Designs, Applications and Challenges

**DOI:** 10.3390/ijms221910698

**Published:** 2021-10-02

**Authors:** Jingyao Li, Fuzhong Zhang

**Affiliations:** 1Department of Energy, Environmental and Chemical Engineering, Washington University in St. Louis, Saint Louis, MO 63130, USA; jingyao.li@wustl.edu; 2Division of Biological & Biomedical Sciences, Washington University in St. Louis, Saint Louis, MO 63130, USA; 3Institute of Materials Science & Engineering, Washington University in St. Louis, Saint Louis, MO 63130, USA

**Keywords:** amyloid fibril, self-assembly, hydrogel, fiber, responsive material, extracellular matrix

## Abstract

Amyloids are self-assembled protein aggregates that take cross-β fibrillar morphology. Although some amyloid proteins are best known for their association with Alzheimer’s and Parkinson’s disease, many other amyloids are found across diverse organisms, from bacteria to humans, and they play vital functional roles. The rigidity, chemical stability, high aspect ratio, and sequence programmability of amyloid fibrils have made them attractive candidates for functional materials with applications in environmental sciences, material engineering, and translational medicines. This review focuses on recent advances in fabricating various types of macroscopic functional amyloid materials. We discuss different design strategies for the fabrication of amyloid hydrogels, high-strength materials, composite materials, responsive materials, extracellular matrix mimics, conductive materials, and catalytic materials.

## 1. Introduction

Proteins and peptides play important functional roles in biology and provide the molecular basis for various biomaterials in nature, from nanoscale cytoskeleton to macroscopic materials [1,2]. The intrinsic biocompatibility and bioactivity of proteins and peptides also render them great potential for various biomedical applications. Recent progress in recombinant technology and synthetic biology has largely advanced protein synthesis, including difficult-to-express proteins, thus providing starting materials for downstream studies [3,4,5,6]. Advances in material science have revealed the sequence-structure-function relationship for multiple types of protein-based biomaterials [7]. While these scientific and technological advances are encouraging, they also highlight the challenge that needs to be addressed before fully realizing the power of protein-based materials: how to control the structure of protein-based materials at multiple length scales?

To control protein structures beyond molecular scale, amyloids become particularly attractive. Amyloids are proteins that can fold into β-sheet and self-assemble to form elongated and unbranched fibrils that are a few nanometers in diameter and up to a few micrometers in length [8]. They were first identified when studying brain tissues of patients with neurodegenerative diseases [9]. Some fibril forms of amyloid proteins were considered to be associated with Alzheimer’s disease (Amyloid β peptide and Tau protein) [10,11,12,13] or Parkinson’s disease (α-Synuclein) [14,15,16,17,18], though the exact mechanisms are still under heated debate in the field [11,15,16,17]. Similar fibrils were later found in a wide range of organisms, and often play various functional roles, including structural scaffold [19,20], catalysis [21,22,23] and functional coatings [24,25]. Some proteins found in human food source such as milk, egg, meat, or wheat can also self-assemble into amyloid fibrils [26]. A wide bioinformatic survey over the genomes of *Escherichia coli*, *Saccharomyces cerevisiae* and *Homo Sapiens* showed that almost all proteins in nature contain at least one short sequence that is capable of aggregation and forming amyloid fibrils, though not all of them will take such conformation [27]. These findings suggest that amyloid structures alone do not necessarily lead to pathogenicity and can potentially be engineered into biocompatible functional materials.

Amyloid fibrils have attractive mechanical properties. Compared with other natural protein fibrils such as actin or tubulin, amyloid fibrils often have significantly higher moduli due to their dense hydrogen bonding networks. Previous studies have shown that amyloid fibrils, at the microscale, can exhibit a Young’s modulus of 3.3 ± 0.4 GPa and a peak tensile strength of 0.6 ± 0.4 GPa [28,29]. The mechanical properties have made amyloid fibrils attractive for structural materials. Amyloid fibrils are also chemically stable. The microfibers have a low depolymerization rate and can maintain their assembled fibril form under environmental changes in pH, temperature, solvents, and salt concentrations. Fibrils assembled from the GNNQQNY peptide, for example, are found to remain stable in 0.5 M NaOH, pure formic acid, or 4 M guanidinium with 2% SDS [30]. Additionally, amyloid fibrils often have a very large aspect ratio, making them suitable for chemical modifications and coating. Finally, properties of amyloid fibrils can be engineered by modifying their amino acid sequences, further expending their applications.

These attractive properties of amyloid fibrils have inspired the development of various artificial materials for applications in sensors [31], medicines [32,33], fabrics [34,35], and other functional materials [36,37,38]. The use of amyloid in nanomaterials has been extensively reviewed previously [20,39,40,41,42,43,44,45]. This review focuses on recent advances in engineering amyloids peptides or proteins as functional materials in macroscopic scales. We highlight the numerous applications such as hydrogels, fibers, composites, sensors, and catalysts.

## 2. Structure and Property of Amyloid Fibrils

When examined by X-ray diffraction, all amyloid fibrils display characteristic cross-β diffraction patterns [46], including one diffuse reflection at 4.8 Å and one at approximately 10 Å [47]. Atomic resolution crystal structures later revealed that these amyloid peptides self-assemble into β-sheet extending along the fibril axis and packed with other β-sheets into steric-zippers through non-covalent interactions between functional groups on side chains [48], which agrees with the basic features highlighted by the X-ray diffraction results. The reflection at 4.8 Å arises from the stackings of β-strands, while the 10 Å spacing arises from the separation between adjacent β-sheets [8,47]. These steric-zippers are assembled from short fibril-forming peptides which are usually 5–8 amino acids long. They form hydrogen bonds between adjacent β-strands along the fibril axis. The side chains of these amino acids can be aromatic, charged, or hydrophobic, thus providing thermodynamically favored interactions between β-sheets [8,49].

Although they share the same diffraction pattern, differences in fibril symmetry exist and were used to categorize amyloid fibrils. The β-strands within amyloid fibrils can be either parallel or antiparallel, and their side chains can be either antifacial (even/odd residues segregated on different sides of the β-sheet) or equifacial (even/odd residues distributed on both sides of the β-sheet). With these two criteria, all amyloid fibrils can be divided into four symmetrical categories: parallel/equifacial, parallel/antifacial, antiparallel/equifacial and antiparallel/antifacial [8,50]. However, parallel/equifacial fibril has not been discovered so far, which can be attributed to: (1) parallel β-sheets are relatively unfavorable compared with antiparallel β-sheets thermodynamically in the proliferation of fibrils due to the hydrogen bonding geometry, and (2) equifacial β-sheets reduce the chance for residues to form a ladder, which could greatly improve the stability of the fibril [51]. In addition to the differences in symmetry, there are also different registries found in different amyloid fibrils [52,53]. A β-sheet is defined as out-of-register when its translational repeat direction is not orthogonal to the strand direction, and amyloids with out-of-register β-sheets are referred to as an out-of-register amyloid, in contrast to in-register amyloids. The structural details of amyloid fibrils can be crucial to their material properties, especially when engineering novel amyloid materials.

Amyloid fibrils are thermodynamically stable due to a collection of non-covalent interactions, including van der Waals forces, π–π stacking, electrostatic forces, hydrogen bonds, and hydrophobic effect occurring among the β-strands and β-sheets [54,55,56,57]. Changes in amyloid peptide sequence can greatly influence both the structure and stability of a fibril [58]. Formation of amyloid fibril is a kinetic driven process. The rate-limiting step in the formation of amyloid fibril is believed to be amyloid seeding [48]. Seeding starts when three to four peptides are in close proximity while simultaneously exposing their zipper-forming segments, which is rare and thus creating a high energy barrier. Once this energy barrier is bypassed, monomers will grow from the seed, one strand at a time. This fibril extension process follows a first-order kinetic model, and fibril can grow rapidly [59]. Multiple environmental factors can affect the self-assembling process, including pH, temperature, solvent, salt concentration, and metal ions. Thus, both thermodynamic and kinetic factors make contributions to the super-molecular assembly of amyloids and are handles for the design of artificial functional amyloid materials.

## 3. Macroscopic Functional Amyloid Materials

### 3.1. Functional Amyloids in Nature

Although the term amyloid is often associated with diseases, scientists have found various cases where amyloid fibrils play important functional roles in nature, from prokaryotic cells to mammals. Amyloids help to adhere to abiotic surfaces [60], detox harmful compounds [61], resist antibiotics [62], direct morphological differentiation of filamentous bacteria [63], and assist electron transport [64].

Many bacteria express and secrete amyloid proteins to the extracellular environment, which, upon self-assembly, form biofilm. In the case of *E. coli*’s curli fibrils, amyloid CsgA proteins assemble on top of the nucleator protein CsgB and form micrometer long fibrils, which further interact to form biofilm [24,65,66]. Similar amyloid fibrils are also discovered in other bacteria species [67,68,69]. In insects, such as silkmoths, amyloids are used to make eggshell for protection [70]. In mammalians, except for their involvement in neurodegenerative diseases, amyloids also play important roles in mitigating the toxicity of melanin formation [23], the formation of long-term memories [71], and the function of pituitary secretory granules [72]. These findings have drastically changed the view over amyloids and demonstrate that they can also play beneficial physiological roles. Molecular level mechanisms on the difference between pathogenic and benign amyloids are still not very well understood. However, it has been proposed that the incompletely self-assembled oligomeric intermediates may be responsible for the cellular toxicity [73]. This will suggest that when beneficial amyloid fibrils are formed, the reaction will go to completion to prevent the accumulation of oligomeric byproducts.

With the discovery of various natural functional amyloids, it is becoming increasingly clear that amyloids can be engineered for real-world applications. In the following section, we will discuss recent progresses in the fabrication of amyloid-based macroscale materials, including hydrogels, macro-fibers, composites, sensors, extracellular matrices, and catalysts.

### 3.2. Artificial Macroscopic Functional Amyloids

#### 3.2.1. Amyloid-Based Hydrogels

Hydrogels are water-containing soft materials with cross-linked polymer networks. Crosslinks between polymer chains can be formed using either covalent bonds or noncovalent interactions. Some proteins can be cross-linked to form hydrogels that have great potentials in biomedicine due to their biocompatibility, biodegradability, and amenable to engineering [74,75]. Various methods have been developed to induce the hydrogelation of proteins and peptides, either through the formation of covalent bonds between peptide chain via pre-modification or through noncovalent interactions such as aromatic stacking or hydrogen bonding [76]. Amyloids are among the prominent candidates for protein hydrogelation because their assembled cross-β fibrils provide extensive intermolecular cross-links.

Natural functional amyloids are among the top candidates to be engineered and fabricated into hydrogels. Two of the most studied are β-lactoglobulin and lysozyme. Both of these proteins are non-toxic food-origin and have high propensity to self-assemble into amyloid fibrils. Several studies have utilized the amyloid form of lysozyme to fabricate antibacterial hydrogels. Hu et al. reported a strategy to form hydrogels by adding polyphenol, such as epigallocatechin gallate (EGCG), to lysozyme amyloid fibrils. Hydrogels were formed after 12 h of incubation at pH 5.8 [77]. The polyphenol served as a crosslinker that interacted with multiple amyloid fibrils through hydrogen bonding, π−π stacking, and hydrophobic interactions. This lysozyme amyloid hydrogel was thermally stable, antibacterial due to the lysogenic activity of lysozyme, and had low cytotoxicity to human cells, therefore providing an attractive candidate for biomedicine [77,78].

Many amyloid fibrils can form hydrogel without additional cross-linker, but through inter-fibril non-covalent interactions including hydrogen bonding, electrostatic forces, π–π interactions, and hydrophobic effects, mediated by amino acid side chains [55,79,80]. Hydrophobic amino acids are usually preferred for this purpose. Medini et al. studied the IKHLSVN peptide sequence from β-interface of peroxiredoxin III [81]. Several residues in this short peptide were mutated to tyrosine, whose side chain allowed fibril-fibril interactions through π−π stacking and hydrogen bonding. These mutants resulted in different fibril patterns, but all displayed amyloid characteristics. Mechanical properties of the hydrogels were also affected by these mutations, suggesting that hydrogels derived from this type of peptides have tunable properties [81]. Identifying short amyloid-forming peptide sequences is useful as it provides building blocks for constructing complex materials. In one example, a tetrapeptide DLII was identified as the shortest fibrillogenic motif from the amyloid TDP-43. DLII and its tetrapeptides variants (YLII, KLII, NLII, and LIID) showed strong hydrogelation propensity at 0.1 wt % [82]. Decandio et al. reported an amyloid-inspired model octapeptide [Arg-Phe]_4_, with arginine promoting hydrogen bond formation and phenylalanine establishing π−π interactions (Figure 1A) [83]. The designed sequence formed long fibrils at 0.17 wt %. At 1 wt %, these fibrils gradually interconnected into a high viscosity gel-like network [83]. Similar design principles have been used to develop other amyloid-like hydrogels using short peptides [84,85] or peptide oligo repeats [86].

Aside from hydrophobic amino acids, the combination of tyrosine and polar amino acids was also used in the design of amyloid hydrogels. The design principle was inspired by natural prion proteins whose core regions usually contain mostly polar residues such as Asn, Gln, Ser, and Gly, as well as Tyr [89]. This sequence pattern has been used to design dynamic and reversible amyloid materials [20,90]. For example, Diaz-Caballero et al. designed four minimalist binary-patterned peptides: NYNYNYN, QYQYQYQ, SYSYSYS, and GYGYGYG (Figure 1B) [87]. Tyrosine residues were used to promote both intra-fibril π−π stacking and inter-fibril dityrosine crosslinks. The polar residues were found to affect the concentration needed for fibril self-assembly [87].

The chirality of amino acids can potentially affect the structure of amyloid fibril, thus providing additional control over hydrogel assembly and its physical and mechanical properties [91]. Marchesan et al. reported hydrogelation of Aβ-derived tripeptides Val-Phe-Phe with the chirality of the central amino acid differing from the other two amino acids (D-L-D and L-D-L in stereochemistry) [92]. The introduction of D-amino acids to tripeptide sequences removed steric clashes between amino acid side chains and thus allowing interdigitation of tripeptide stacks into zippers. Similarly, Kralj et al. used diphenylalanine with different chirality (^D^Phe-^L^Phe) to controll fibril structures and successfully obtained hydrogels [88]. In contrast to homochiral diphenylalanine (e.g., ^L^Phe-^L^Phe), the heterochiral ^D^Phe-^L^Phe fibril had homogenous size distribution and reduced cytotoxicity (Figure 1C) [88]. These examples demonstrate the potential of heterochirality as a strategy to design amyloid hydrogels [91].

Recently, a study carried out by Bal et al. proposed an interesting strategy called non-equilibrium amyloid polymerization [93]. While the sequence KLVFFAE does not form hydrogel on its own, the addition of a small amount of KLVFFAL, which contains a hydrophobic leucine, facilitated the nucleation core formation and promoted peptide aggregation, leading to hydrogelation. The authors designed a peptide HLVFFAE-NP, where E-NP stands for 4-nitro phenol (NP) functionalized glutamic acid. The new sequence formed fibrils similar to that of KLVFFAE. Upon fibril formation, histidine sidechains in the fibril displayed esterase activity, which removed NP from glutamic acid residue and destabilized the fibril. This created a negative feedback cycle where fibril formation catalyzed the depolymerization of fibrils after their formation. On a macroscale, the peptides formed hydrogels in 1 h and became fluid after 5 h [93].

#### 3.2.2. High-Strength Materials

The attractive mechanical property of amyloid at nanoscale has motivated decades of research to translate such properties into macroscopic materials. The most common types of high-strength amyloid materials engineered so far are free-standing films and fibers.

A free-standing film can be made by allowing amyloid fibrils to elongate under conditions that favor intermolecular interactions. Plasticizers are added at the end of fibril growth phase to start hydrogelation. The hydrogel is then dried on a flat polytetrafluoroethylene film to form a free-standing film. Using this method, films were made from either β-lactoglobulin or lysozyme [94]. Both films exhibited Young’s modulus ranging from 5.2 to 7.2 GPa, close to that of amyloid fibrils at nanoscale (2–19 GPa) [29,54,95,96].

Spider silk fiber is one of the strongest and toughest macroscopic fiber materials in nature. As with amyloid aggregates, natural spider silk fibers are rich in β-sheet crystallites formed by polyalanine repeat sequences. However, the β-strands in these β-nanocrystals are aligned in parallel with the fiber axis, distinct from that of amyloid fibrils [97,98]. Inspired by the semi-crystalline structure of spider silk fiber, polymeric amyloid proteins were recently designed by combining amino acid sequences from both spider silk protein and amyloid peptides (Figure 2) [99]. These proteins were then spun into macroscopic fibers using a wet-spinning process. The high self-assembly tendency of amyloid peptide promoted the formation of β-nanocrystals during fiber spinning, resulting in enhanced crystallinity, which are known to be critical to fiber strength [99]. Fibers of all tested polymeric amyloid proteins have displayed higher ultimate strength than recombinant spider silk fibers under similar molecular weight. The reduced sequence redundancy (e.g., from AAAAA to FGAILSS) also improved genetic stability and facilitated protein biosynthesis in heterologous host. As a result, a high-molecular-weight (378 kDa) polymeric amyloid protein containing 128 repeating FGAILSS sequences was biosynthesized and spun into fibers. The fibers displayed a high tensile strength (0.96 GPa) and high toughness (160 MJ/m^3^), exceeding most recombinant spider silks and even some natural spider silk fibers [35,100,101]. Similar design rules can potentially be applied to other amyloid sequences, thus drastically expanding the diversity of mechanical-demanding protein-based materials [34].

#### 3.2.3. Amyloid-Inorganic Hybrid Composite Materials

Amyloid fibrils have high aspect ratios and generally display multiple binding sites for small molecules along their surface, which makes them promising templates for the design of organic-inorganic hybrid nanomaterials through specific post-assembly modification or co-assembly approaches. Fabrication of nanoscale amyloid-inorganic hybrid materials has been previously reviewed [102]. Here, we focus on the macroscopic composite materials.

In one of the pioneering studies, β-lactoglobulin amyloid fibrils were used as templates for gold deposition, forming gold aerogels [103]. Amyloid peptides were later used in several papers for bio-mineralization. Zhang et al. reported a hybrid β-lactoglobulin/ZrO_2_ (CAF-Zr) membrane that exhibited selective removal of fluoride from water [104]. The charged amyloid scaffold provided anchorage for ZrO_2_ nanoparticles so that these nanoparticles (approximately 10 nm in diameter) can avoid aggregation, thus increasing surface/volume ratio of ZrO_2_ particles by several hundred-fold from those particles without attaching to the scaffold. The hybrid nanocolloids were entrapped with activated carbon, and the composite materials were used to remove fluoride from contaminated water with 99.5% removal efficiency [104]. A similar strategy was also used to improve the catalytic efficiency of metal nanoparticles by increasing their aspect ratio. β-lactoglobulin fibrils decorated with gold and palladium nanoparticles that catalyze the reduction of 4-nitrophenol to 4-aminophenol were fabricated into membrane through vacuum filtration. The membrane can achieve complete catalytic conversion of this reaction within one single passage through [105]. Bolisetty et al. presented a β-lactoglobulin–carbon hybrid membranes made from low-quality whey protein that showed high efficiency in the removal of heavy metal ions, arsenite/arsenate, and clinically relevant radioactive compounds from hospital wastewater [106,107,108]. Cao et al. reported stiff fibril-silica composite gels using β-lactoglobulin fibrils. Fibrils were mixed with silica precursor tetraethyl orthosilicate to direct silification on the surface of amyloid fibrils [109]. Similarly, Ha et al. designed a scaffold for hydroxyapatite (HA) crystal growth using a lysozyme nanofilm coating [110]. The lysozyme nanofilm contained multiple carboxyl and hydroxyl groups that chelate Ca^2+^ ion. After incubation in 0.02 M CaCl_2_ solution, the nanofilm was transferred into simulated body fluid to form a HA layer. The authors demonstrated that this strategy allowed HA integration on various surfaces, including ceramics, metals, semiconductors, and synthetic polymers irrespective of their size and morphology [110]. Li et al. designed a peptide, consisting of a LLVFGAK amyloid motif and a MLPHHGA mineralization motif [111]. The mineralization motif contained numerous functional groups, for sequestering HA from simulated body fluid. After self-assembly, the nanosheets were conjugated with 3D-graphene foam (GF) to form 3D GF-PNSs hybrid scaffolds for HA sequestration. The resulting 3D-GF-PNSs-HA minerals exhibited adjustable shape, super low weight, high porosity, and excellent biocompatibility, proving potential applications in both bone tissue engineering and biomedical engineering [111].

Bacterial amyloids such as curli have also been employed to direct biomineralization, where mineral-binding peptides were integrated into the cuili subunit CsgA sequence. Yang et al. reported a hybrid protein CsgA-DDDEEK as a coating material capable of sequestering HA [112]. The DDDEEK sequence originates from salivary acquired pellicles in the dental plaque biofilm and has a strong ability to absorb mineral ions and induce the formation of biominerals. The engineered biofilm was used to coat Ti_6_Al_4_V nanoparticles. The resulting composite exhibited long-term stability, improved force-resisting capacity, and high biocompatibility without triggering immune responses in animal experiments, making it a potential candidate for implant materials [112]. An alternative approach was developed by substituting three of the five pseudorepeats (R2-R4) of CsgA to HA-binding peptide, resulting in a R1-HAP-R5 mutant CsgA (Figure 3). The remaining pseudorepeats R1 and R5 were sufficient for fibril formation. When incubating with CaCl_2_ or Na_2_HPO_4_, HA-derived crystals were able to form uniform and dense nanosized minerals in the R1-HA-R5 hydrogel [113].

#### 3.2.4. Responsive Materials for Sensing

Responsiveness to external stimuli is not an intrinsic feature of amyloid fibrils. However, by tuning the interactions between fibrils or by introducing functional components to the system, responsive and sensory amyloid materials can be fabricated. Here, we summarize recent macroscopic amyloid materials whose morphology, mechanical properties, or color change in response to pH, temperature, salt concentration, magnetic field, or other environmental stimuli.

Multiple strategies have been developed to fabricate pH-responsive amyloid hydrogels. One straightforward method is to immobilize existing pH-responsive molecules to amyloid hydrogels. Saldanha et al. fused pHuji, a pH-responsive fluorescent protein, to CsgA [114]. The fusion protein was made into textiles via vacuum filtration. The resulting materials were mechanically stable and underwent yellow to red transition upon a pH increase [114]. pH change can also be used to control hydrogel mechanical properties via pH-sensitive interactions between amyloid and additives. Li et al. fabricated a pH-responsive hydrogel by mixing β-lactoglobulin fibrils with sulfonated multi-walled carbon nanotube (MWCN) [115]. At pH 2, the negatively charged MWCN interacted with the positively charged β-lactoglobulin fibrils and thus formed a hydrogel. When the pH was increased to 7, higher than the pI of β-lactoglobulin, the amyloid fibrils became negatively charged, thus disassociate from MWCN, leading to a 100-fold decrease in hydrogel modulus [115].

Responsiveness to heat has also been achieved in amyloid hydrogels. Ozbas et al. synthesized a 20-residue β-hairpin peptide VKVKVKVKVPPTKVKVKVKV, where VK repeats formed β-sheet, and PP formed a type II turn, packing the hydrophobic Val sidechains between two β-sheets [116]. Hydrogelation occurred upon the addition of salt to neutralize the repulsion between positively charged peptides. Rheology tests revealed that hydrogel modulus also responded to temperature. A higher temperature at 37 °C promoted the formation of β-sheets, making the hydrogel stiffer compared to that at room temperature [116].

Amyloid-based scaffolds can be used to hold magnetic Fe_3_O_4_ nanoparticles, which allows the hydrogel to be magnetically sensitive. Bolisetty et al. prepared a β-lactoglobulin-Fe_3_O_4_ composite hydrogel, where the amyloid fibrils can be aligned via their associated Fe_3_O_4_ particles in three dimensions using magnetic fields as low as 0.1 T (Figure 4) [117]. The hydrogel further underwent a reversible sol-gel transition as magnetic field increased to 1.1T [117]. Lutz-Bueno et al. reported water-responsive wires made from gelatin and β-lactoglobulin fibrils to mimic self-winding of plants’ tendrils [118]. The mechanism of this self-winding behavior relied on the relatively low alignment of amyloid fibrils within the wire, which triggered the rotatory force in water and thus transferred the chirality of gelatin and amyloid fibrils across multiple length scales. The water-responsive wires were then incubated with magnetic Fe_3_O_4_ nanoparticles, allowing the wire to elongate linearly with increasing magnetic field strength. Such materials can potential be used as underwater stretchable sensors, enabling new applications [118].

#### 3.2.5. Extracellular Matrix (ECM) to Sustain Viable Cells

Due to their connections to neurodegenerative diseases [119,120,121], amyloid materials were historically not considered to be suitable for biomedical applications. This view was later changed as more evidence from recent studies suggested that amyloids may only be a nontoxic byproduct, and some amyloids were found to be functional. Now, attracted by their mechanical properties, increasing research efforts have explored amyloid materials in artificial extracellular matrices (ECMs). Although natural amyloids often do not contain cell adhesion motifs (e.g., RGD from fibronectin), functional peptide sequences can be added to amyloid proteins by genetic fusion and bioconjugation.

Reynold et al. used a lysozyme fibril network as artificial ECM to support the growth of fibroblast or epithelial cell lines [122]. Residues 66–68 within lysozyme contains a tripeptide DGR that is similar to RGD. This minimal fibronection-binding domain promoted integrin-mediated focal adhesions [123,124,125]. Cells remained viable for up to 7 days on the fibrillized lysozyme matrix with increased focal adhesion compared to monomeric lysozyme. Similarly, the amyloid peptide IKVAV from laminin-1 was incorporated into an elastin-like polypeptide (ELP). This amyloid-ELP hybrid protein formed a hydrogel upon crosslinking by multi-arm thiol- and acrylate-functionalized polyethylene glycol (PEG) crosslinker. ThT assay confirmed the presence of amyloid fibrils formed from the IKVAV sequence. Both in vitro and in vivo experiments demonstrated that the hydrogel promoted the growth of sensory neuron without noticeable cytotoxicity and inflammatory effects [126]. Deidda et al. fused the amyloid peptide NSGAITIG from adenovirus fiber shaft to the RGD motif [127]. A cysteine residue was incorporated for potential bioconjugation. The assembled peptide hydrogel supported both adhesion and proliferation of fibroblast NIH/3T3 cells, suggesting promising biomedical applications [127]. Additionally, purified inclusion body containing both human chaperone Hsp70 and unfused human FGF-2 were used to form scaffold. Proteins in this inclusion body adopted amyloid-like cross-β structures. The scaffold displayed good cell growth properties [128].

Stem cell transplantation has attracted an increasing amount of attention as a potential treatment for multiple diseases and traumas including cancers, cardiac disease, stroke, Alzheimer’s disease, and severe burns. It is critical to precisely control the microenvironment of stem cells as it affects cell differentiate. So far, effective stem cell transplantation has proven to be extremely challenging in clinical tests because stem cells often lose viability quickly after transplantation [129,130] or drift away from the targeted lesion [131]. Consequently, researchers have been focusing on developing artificial ECM to encapsulate, sustain and direct the differentiation of stem cells. Several amyloid-based ECMs have been presented as candidates for such vital tasks. Das et al. reported an α-synuclein-inspired hydrogel that promoted stem cell differentiation to neurons [132]. The self-recognition motif of α-synuclein VTAVA has the highest β-propensity within the protein and was chosen. The two alanine residues at the third and the fifth position of the peptide were mutated into more hydrophobic residues to improve the extent of crosslinking. The Thr at the second position was mutated to Tyr or His to provide extra π−π stacking or sidechain ionization (Figure 5). Hydrogels fabricated from these peptides not only provided mechanical support to cells, but also directed cell differentiation [133]. Implantation of the hydrogel into mouse brain did not trigger severe immune responses. In vitro assays showed that the hydrogel supported neuronal differentiation, possibly via mechanical stimulation [132]. Jacob et al. designed a series of peptides based on the aggregation-prone Aβ42 protein [38]. Hydrogelation occurred after a heat/cool cycle and the mechanical property was tunable through modulation of peptide concentration and salt concentration. The hydrogel was used as a scaffold for stem cell differentiation and kept the entrapped cells viable for more than 48 h [38].

#### 3.2.6. Conductive Materials

Benefitting from their fibrillar morphology, amyloid fibrils can be engineered or modified into conductive materials. With a rational design, high electron mobility can be achieved along the highly ordered protein fibrils, resulting in conductive wires and gels that are also biocompatible and biodegradable.

One strategy to improve electron mobility along amyloid fibrils is the incorporation of conductive nanoparticles. Han et al. demonstrated that β-lactoglobulin fibrils can template the polymerization of pyrrole into conductive polypyrrole [134]. After pyrrole polymerization, the mixture can be readily fabricated into mechanically stable aerogel with lyophilization. The conductivity of the aerogel responded to environmental stimuli such as pressure with high sensitivity, making it an ideal candidate for wearable biosensors [134].

Some amyloids consist of multiple aromatic amino acids in their sequences to help initiate fibrilization and stabilize their self-assembled structure. These aromatic rings, when properly aligned, can serve as the structural basis of delocalized π clouds and become conductive over a long range [135]. The conductive pili of Geobacter bacteria, in particular, has attracted attention. Over the years, researchers have established efficient protocols to collect and produce engineered curli nanofibrils fused with different tags or with designed mutations [136,137]. Kalyoncu et al. fabricated conductive biofilm by introducing aromatic amino acids to the curli sequence to create delocalized π clouds similar to that in conductive pili of Geobacter bacteria [138]. It was later found that adding tri-tyrosine or tri-tryptophan peptide to CsgA can effectively improve biofilm conductivity [139]. The idea of engineering delocalized π clouds was further proved by computationally designed CsgA mutants carrying aromatic residues for delocalized electron transport across the curli fibril [140].

#### 3.2.7. Catalytic Materials

Amyloid fibrils can be engineered to form unique chemical environments to catalyze reactions, where catalytic sites can be introduced to the amyloid sequences and the fibril structure provides a critical microenvironment for reactions to occur.

Histidine often plays an important roles in esterase, serving as both proton donor or acceptor during ester hydrolysis [141]. Spatially organizing multiple histidine side chains into close proximity by controlled amyloid assembly can promote esterase activity. Diaz-Caballero et al. designed such a peptide using only histidine and tyrosine (HYHYHYHY), which self-assembled into cross-β fibrils (Figure 6) [142]. Tyrosine residues promoted the self-assembly of the peptide into nanofibrils while also displaying oxidase activity. Histidine was chosen for its propensity to form non-covalent interactions between fibrils and for its hydrolase activities [87,143,144]. Hydrogelation occurred at pH 8.0, and reversibly turned to fluid at pH 4.0. The pH-responsiveness, together with the high esterase-oxidase dual enzyme activity, has opened up new applications for amyloid fibrils as catalytic materials [142].

Garcia et al. took one step further and designed a tripeptide His-^D^Phe-^D^Phe that formed a thermally reversible hydrogel with esterase activity [145]. These hydrogels went through a sol-gel transition temperature at 45°C under neutral pH [145]. Carlomagno et al. designed an amyloid-like octapeptide-based hydrogel with catalytic activities. The His–Leu–^D^Leu–Ile–His–Leu–^D^Leu–Ile octapeptide relied on the hydrophobicity of leucine and isoleucine to self-assemble and was found to form hydrogel at 10 mM, which is much lower compared to previously reported tripeptide [146].

## 4. Perspectives and Challenges

In this review, we discussed recent efforts in engineering amyloid peptides or proteins as macroscopic functional materials. The strong mechanical properties and chemical stability of amyloids have been harnessed to produce strong macroscopic materials. Their sequence programmability has enabled engineering of amyloid fibrils for tunable properties, stimuli-responsiveness, and various functions. Their biocompatibility has opened the possibility for future biomedical applications. Overall, a wide range of amyloid proteins and peptides have been used to fabricate membranes, fibers, hydrogels, composites, and catalysts. To date, some amyloid-based materials have been successfully produced in industrial settings by private companies and are ready for commercialization [107,108,147].

While progresses have been made, challenges remain. Firstly, coherent and predictable design rules for macroscopic amyloid materials have not yet been established for multiple types of materials. When designing amyloid hydrogels, several strategies exist, such as using aromatic or hydrophobic residues to promote non-covalent interactions. However, these strategies are largely empirical and not robust. Addition of new sequences may lead to unexpected results. Because interactions between amyloid fibrils are heterogeneous, accurate prediction on the properties of macroscopic materials from protein sequence remains challenging even with the help of powerful computational tools such as AlphaFold2 [148]. Secondly, amyloid proteins with new functionalities are needed to further expand their applications. This can be potentially achieved by carefully fusing new functional peptides or proteins into fibril-forming amyloid sequences, so that the fusion protein can be assembled in amyloid fibrils while maintaining the activity of the fused functional protein. Thirdly, scalable and cost-effective production of amyloid peptides is a serious issue for high-quantity applications. One possible solution is to use synthetic biological strategies to produce these polypeptides from engineered microbes. With continued development, we believe amyloid materials will become more popular in material engineering and translational medicine.

## Figures and Tables

**Figure 1 ijms-22-10698-f001:**
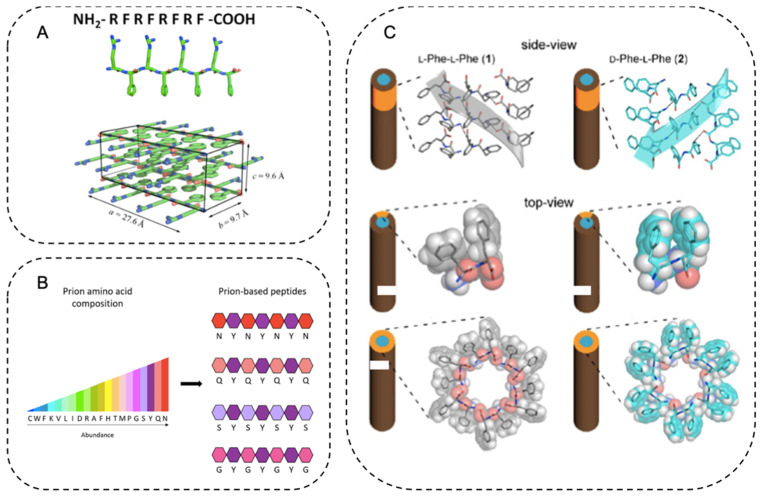
Strategies for the fabrication of amyloid-based hydrogels. (**A**) β-sheets of the designed octapeptide [Arg-Phe]_4_ packed through extensive π−π stacking, while the arginine residues point to the outside, ready to form inter-fibril hydrogen bonds. Reprinted with permission [83]. Copyright 2015, American Chemical Society. (**B**) Prion-inspired peptides consist of tyrosine residues and other polar amino acids. Tyrosine residues contributed to the self-assembly with π−π stacking, while the rest of residues provided the polar context and weak interactions. Reprinted with permission [87]. Copyright 2018, American Chemical Society. (**C**) Altered self-assembly patterns in homochiral and heterochiral diphenylalanine. Different chiral patterns led to different hydrogel physical properties in macroscale. Reprinted with permission [88]. Copyright 2020, American Chemical Society.

**Figure 2 ijms-22-10698-f002:**
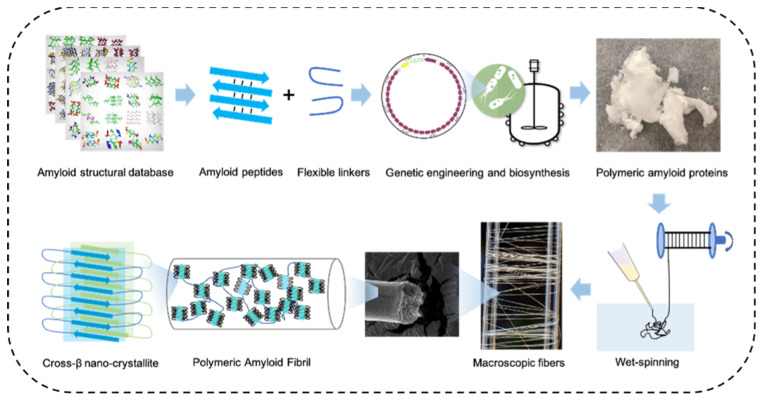
Fabrication procedures of mechanically strong macroscopic polymeric amyloid fibers. Amyloid peptides were incorporated into tandem repeats of spidroins to replace the original polyalanine segments. Upon spinning, these amyloid peptides promoted the formation of β-nanocrystals, consequently making the macroscopic fiber stronger. Reprinted with permission [34]. Copyright 2021, American Chemical Society.

**Figure 3 ijms-22-10698-f003:**
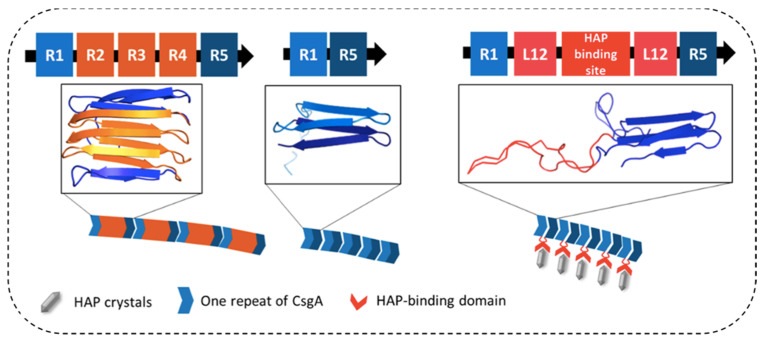
Engineered *E. coli* curli protein to direct the biomineralization of hydroxyapatite. The pseudorepeats in the center of CsgA were substituted with hydroxyapatite-binding sequences. The remaining pseudorepeats R1 and R5 were sufficient to form fibrils, which directed the biomineralization of hydroxyapatite via the fused HAP-binding domain. Reprinted with permission [113]. Copyright 2020, American Chemical Society.

**Figure 4 ijms-22-10698-f004:**
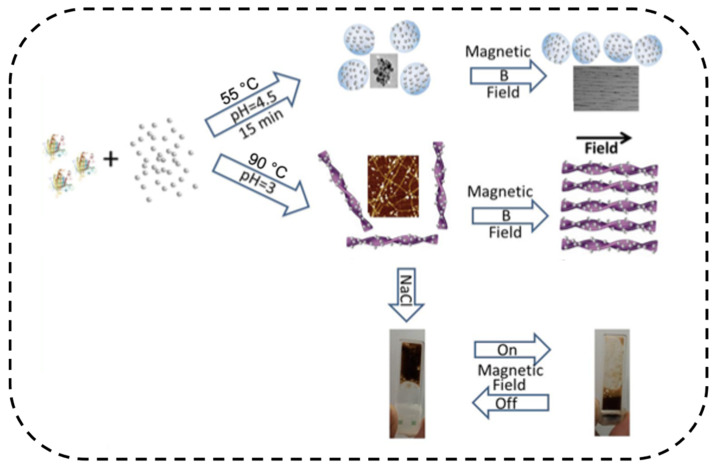
Magnetic-responsive β-lactoglobulin-Fe_3_O_4_ hybrid hydrogel. When a magnetic field was applied, amyloid fibrils became aligned. As a result, the amount of intermolecular contact reduced, and the hydrogel collapsed. When the magnetic field was turned off, hydrogel formed again. Reprinted with permission [117]. Copyright 2013, American Chemical Society.

**Figure 5 ijms-22-10698-f005:**
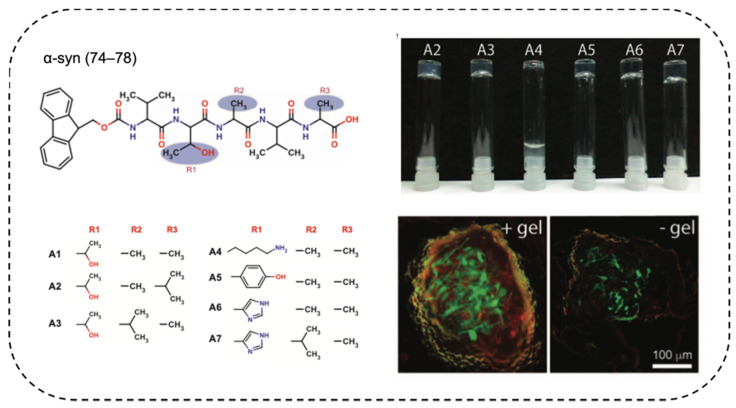
α-synuclein-inspired hydrogel that supported neuron differentiation. The β-sheet-zipper-forming sequence of α-synuclein VTAVA was mutated to facilitate the self-assembly of the peptides, which provided mechanical support, and enabled the proper differentiation of stem cells into neurons. Reprinted with permission [132]. Copyright 2016, Nature Publishing Group.

**Figure 6 ijms-22-10698-f006:**
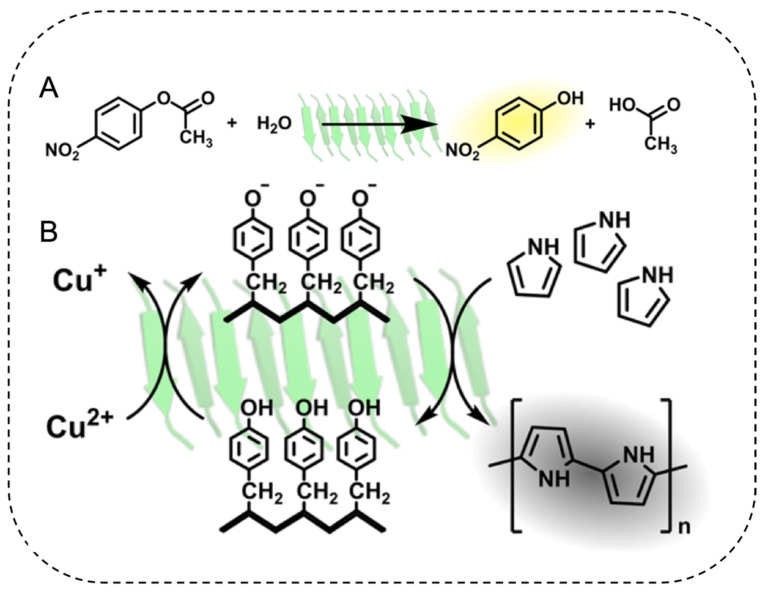
Design of amyloid fibrils with dual esterase-oxidase activities. The self-assembled HYHYHYHY fibrils provided important microenvironments for histidine and tyrosine sidechain to exhibit esterase (A) and oxidase (B) activities, respectively. Reprinted with permission [142]. Copyright 2020, American Chemical Society.

## Data Availability

No new data were created or analyzed in this study. Data sharing is not applicable to this article.

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
