# Peer review of "Amyloids as Building Blocks for Macroscopic Functional Materials: Designs, Applications and Challenges"

_ijms, 2021, doi:10.3390/ijms221910698_

Round 1

Reviewer 1 Report

Zhang et.al thoroughly reviewed the recent advances in engineering amyloids peptides or proteins as functional materials in macroscopic scales. Personally, I enjoyed reading the summary of all the recent development in amyloid engineering for the functional materials. Eventhough authors did their best in review preparation, I suggest to consider below points for futher improving the review articles.

1, Amyloids are always considered as toxic. It is beeter to include a short paragraph describing that food grade amyloid for further applications citing the article “Food protein amyloid fibrils: Origin, structure, formation, characterization, applications and health implications”, Advances in colloid and interface science, 2019

2, amyloid functional materials are not only produced on the lab scale, it is also scale up to industrial scale production for the variety of applications. Several companies were started with the amyloid based material for the industrial production and products were sold in the market.  It is good mention in the review with short paragraph explain the real scale existing applications. For example “Hi-aspect” produce amyloid gels for the personal care products and wound healing applications. “BluAct Technologies “ produces amyloid based membranes for the waste water and drinking water purification. “Goold Ag” produces amyloid based gold for the jewellery and conducting plating applications.

3, amyloid based conductive materials can be expanded a little more citing the “Amyloid Fibril‐Templated High‐Performance Conductive Aerogels with Sensing Properties”, & “Amyloid directed synthesis of titanium dioxide nanowires and their applications in hybrid photovoltaic devices”.

4, in the catalysis part amyloid based hybrid materials have high potential for variety of the catalytic applications “Hybrid amyloid membranes for continuous flow catalysis”

5, authors need to cross check citation format and English for the entire manuscript.

Author Response

Dear Reviewers:

Please note that the line numbers in this response correspond to the marked copy of the manuscript we submitted as the supplementary material for review.

Reviewer 1:

Zhang et.al thoroughly reviewed the recent advances in engineering amyloids peptides or proteins as functional materials in macroscopic scales. Personally, I enjoyed reading the summary of all the recent development in amyloid engineering for the functional materials. Eventhough authors did their best in review preparation, I suggest to consider below points for futher improving the review articles.

Response: We thank the reviewer for the positive review of our manuscript, and we have addressed the reviewer’s concerns below point-by-point.

  1. Amyloids are always considered as toxic. It is beeter to include a short paragraph describing that food grade amyloid for further applications citing the article “Food protein amyloid fibrils: Origin, structure, formation, characterization, applications and health implications”, Advances in colloid and interface science, 2019

Response: We thank the reviewer for the suggestion. We have included related discussion and citation in our revised manuscript. Please find the changes in 42-51.

  1. Amyloid functional materials are not only produced on the lab scale, it is also scale up to industrial scale production for the variety of applications. Several companies were started with the amyloid based material for the industrial production and products were sold in the market. It is good mention in the review with short paragraph explain the real scale existing applications. For example “Hi-aspect” produce amyloid gels for the personal care products and wound healing applications. “BluAct Technologies “ produces amyloid based membranes for the waste water and drinking water purification. “Goold Ag” produces amyloid based gold for the jewellery and conducting plating applications.

Response: We appreciate the information that the reviewer provided. We have included discussions on the current commercialization of amyloid-based macroscopic materials. Please find these discussions in lines 499-500. We decided not to mention the names of these private companies, but cited the papers on what their technologies are based.

  1. Amyloid based conductive materials can be expanded a little more citing the “Amyloid Fibril‐Templated High‐Performance Conductive Aerogels with Sensing Properties”, & “Amyloid directed synthesis of titanium dioxide nanowires and their applications in hybrid photovoltaic devices”.

Response: We thank the reviewer for the suggestion. We have included these important papers in the manuscript. Please find these changes in line 437-447.

  1. In the catalysis part amyloid based hybrid materials have high potential for variety of the catalytic applications “Hybrid amyloid membranes for continuous flow catalysis”

Response: We thank the reviewer for suggesting this paper. We have included this paper in the “Amyloid-inorganic hybrid composite materials” section of the revised manuscript. Please see lines 288-295.

  1. authors need to cross check citation format and English for the entire manuscript.

Response: We have cross-checked our citation format and language. We have also made changes to avoid confusions. Please find these changes in line 83, 108, 155 and 502.

Reviewer 2 Report

The review article by Li and Zhu is a literature survey of the state of the art on the use of amyloid-like self-assembling peptides/proteins to generate new functional materials. After a general introduction on the basic structural features of amyloid-like assemblies, the review specifically focuses on amyloid-based hydrogels, fibers and composite materials as well as on materials that can be exploited as sensors and catalysts.

I found the review of interest and clearly written. I believe that it can be considered for publication. However, I have some points that should be addressed upon revision.

1) Although at lines 57-60 the authors cite some previous reviews on this topic and also specify the main goal of their work, I believe that this part should be expanded by carrying out a more exhaustive summary of the literature reviews present in literature. The authors may consider for example these papers: Hamley Chem Rev DOI: 10.1021/acs.chemrev.7b00522; Accardo et al. Front Bioeng Biotechnol doi: 10.3389/fbioe.2021.641372; Shimanovich et al Adv Mater doi: 10.1002/adma.201706462; Maji et al. Biomacromolecules doi: 10.1021/acs.biomac.8b00279;)

2) The authors define Amyloids as proteins (see for example the beginning of the abstract “Amyloids are proteins that can assemble into cross-β fibrils”). Since a given protein may or not form amyloids depending on its structural state, I would instead define Amyloids as protein aggregates (or deposits/assemblies).

3) The text should be revised to correct typos and to improve a bit its clarity. For example:

Line 73 axe should be axis

Line 71 The term “crystals” is inappropriate as the authors are discussing spots (4.8 and 10 Angstrom) that are typical of the fiber diffraction pattern. The authors should clarify that the atomic level information derived from single crystal analyses is fully compatible with the basic features highlighted by the fiber diffraction.

Line 99 stars should be starts

Line 145 hydron should be hydrogen

Line 235-237 The authors should make clear that, although rich in beta-structure, natural silk fibers are quite distinct from amyloid-like aggregates as in the former the fiber axis is coincident with the polypeptide chain. They are not cross-beta.

In several cases, the authors use in the text the expression “Author et al.” without adding the related reference. For clarity, the citation number should be reported in the same sentence.

Author Response

Dear Reviewers:

Please note that the line numbers in this response correspond to the marked copy of the manuscript we submitted as the supplementary material for review.

Reviewer 2:

The review article by Li and Zhu is a literature survey of the state of the art on the use of amyloid-like self-assembling peptides/proteins to generate new functional materials. After a general introduction on the basic structural features of amyloid-like assemblies, the review specifically focuses on amyloid-based hydrogels, fibers and composite materials as well as on materials that can be exploited as sensors and catalysts.

I found the review of interest and clearly written. I believe that it can be considered for publication. However, I have some points that should be addressed upon revision.

Response: We thank the reviewer’s positive review of our manuscript, and we have addressed the reviewer’s concerns below point-by-point.

  1. Although at lines 57-60 the authors cite some previous reviews on this topic and also specify the main goal of their work, I believe that this part should be expanded by carrying out a more exhaustive summary of the literature reviews present in literature. The authors may consider for example these papers: Hamley Chem Rev DOI: 10.1021/acs.chemrev.7b00522; Accardo et al. Front Bioeng Biotechnol doi: 10.3389/fbioe.2021.641372; Shimanovich et al Adv Mater doi: 10.1002/adma.201706462; Maji et al. Biomacromolecules doi: 10.1021/acs.biomac.8b00279;)

Response: We thank the reviewer for the suggestion. We have included more literatures reviews on this topic, including the papers that the reviewer mentioned. Please find the changes in line 68 of the revised manuscript.

  1. The authors define Amyloids as proteins (see for example the beginning of the abstract “Amyloids are proteins that can assemble into cross-β fibrils”). Since a given protein may or not form amyloids depending on its structural state, I would instead define Amyloids as protein aggregates (or deposits/assemblies).

Response: We thank the reviewer for pointing out this. We have revised the manuscript accordingly to clarify such confusion. Please find changes in line 11.

  1. The text should be revised to correct typos and to improve a bit its clarity. For example:

Line 73 axe should be axis

Line 71 The term “crystals” is inappropriate as the authors are discussing spots (4.8 and 10 Angstrom) that are typical of the fiber diffraction pattern. The authors should clarify that the atomic level information derived from single crystal analyses is fully compatible with the basic features highlighted by the fiber diffraction.

Line 99 stars should be starts

Line 145 hydron should be hydrogen

Line 235-237 The authors should make clear that, although rich in beta-structure, natural silk fibers are quite distinct from amyloid-like aggregates as in the former the fiber axis is coincident with the polypeptide chain. They are not cross-beta.

In several cases, the authors use in the text the expression “Author et al.” without adding the related reference. For clarity, the citation number should be reported in the same sentence.

Response: We thank the reviewer for pointing out these issues. We have revised the manuscript according to the reviewer’s suggestions. Please find these changes in line 83, line 79-81, line 108, line 155 and line 246-248. We have also added citations to sentences with “Author et al.”

Reviewer 3 Report

I have read the current Review manuscript by Li & Zhang on “Amyloids as Building Blocks for Macroscopic Functional Materials: Designs, Applications and Challenges” with great interest. Overall, I find the manuscript well-written, and with potential interest particularly for the nanomaterial’s field, but to some extent also for the neuroscience field. Nevertheless, there are some issues that the authors should address in the revised manuscript.

Although the authors mention the association of amyloid proteins with Alzheimer’s (AD) and Parkinson’s disease (PD) in the Abstract, they fail to further elaborate on that in the manuscript’s main text, which reflects some lack of overall coherence. This is the manuscript's main flaw since i) these amyloidosis are associated with the most common forms of CNS neurodegenerative diseases, with a significant social an economic impact worldwide, and ii) you can not really review about amyloids without further elaborating on the biochemistry and pathophysiology of some of the most important human aggregation-prone proteins e.g. alpha-synuclein.

Therefore, I would strongly suggest the authors to comment on some of the most recent and exciting research happening in the PD field. It is becoming clear that pathological alpha-synuclein, which is the main constituent of Lewy pathology and the hallmark with PD, can propagate from neuron-to-neuron in a templated-assisted manner from the periphery to the central nervous system (PMID: 33632316; PMID: 33978813; PMID: 31254094), contradicting the traditional view of PF as solely a brain disease. The self-perpetuating pathological polymerisation mechanisms by which alpha-synuclein aggregates or “seeds” recruit native alpha-synuclein monomers in infected neurons has become the focus of intense research in recent years and hold the potential for innovative therapies for the PD. The authors should cite these papers and briefly comment on them in the Introduction section.

Author Response

Dear Reviewers:

Please note that the line numbers in this response correspond to the marked copy of the manuscript we submitted as the supplementary material for review.

Reviewer 3:

I have read the current Review manuscript by Li & Zhang on “Amyloids as Building Blocks for Macroscopic Functional Materials: Designs, Applications and Challenges” with great interest. Overall, I find the manuscript well-written, and with potential interest particularly for the nanomaterial’s field, but to some extent also for the neuroscience field. Nevertheless, there are some issues that the authors should address in the revised manuscript.

Although the authors mention the association of amyloid proteins with Alzheimer’s (AD) and Parkinson’s disease (PD) in the Abstract, they fail to further elaborate on that in the manuscript’s main text, which reflects some lack of overall coherence. This is the manuscript's main flaw since i) these amyloidosis are associated with the most common forms of CNS neurodegenerative diseases, with a significant social an economic impact worldwide, and ii) you can not really review about amyloids without further elaborating on the biochemistry and pathophysiology of some of the most important human aggregation-prone proteins e.g. alpha-synuclein.

Therefore, I would strongly suggest the authors to comment on some of the most recent and exciting research happening in the PD field. It is becoming clear that pathological alpha-synuclein, which is the main constituent of Lewy pathology and the hallmark with PD, can propagate from neuron-to-neuron in a templated-assisted manner from the periphery to the central nervous system (PMID: 33632316; PMID: 33978813; PMID: 31254094), contradicting the traditional view of PF as solely a brain disease. The self-perpetuating pathological polymerisation mechanisms by which alpha-synuclein aggregates or “seeds” recruit native alpha-synuclein monomers in infected neurons has become the focus of intense research in recent years and hold the potential for innovative therapies for the PD. The authors should cite these papers and briefly comment on them in the Introduction section.

Response: We thank the reviewer for the suggestions. Pathogenicity and recent progresses on Parkinson’s disease are important aspects of amyloid research, and we have briefly comment on it in the revised manuscript as the reviewer suggested. We have also included more citations and comments in the introduction, including the papers the reviewer had recommended. Please find these changes in line 39-42. Because the focus of this review is mostly on the material applications of amyloids rather than their involvement in disease, we prefer to keep the discussion of PD concise.

Round 2

Reviewer 3 Report

The authors have successfully addressed my concerns, so I hereby recommend the revised version for publication.